

# The secret life of deep-sea shrimps: ecological and evolutionary clues from the larval description of *Systellaspis debilis* (Caridea: Oplophoridae)

Cátia Bartilotti[1,*] and Antonina Dos Santos[1,2,*]

[1] IPMA, Lisboa, Portugal
[2] CIIMAR, Porto, Portugal
* These authors contributed equally to this work.

## ABSTRACT

Currently there are 21 shrimp species in the northeastern Atlantic and Mediterranean Sea which are considered to belong to the superfamily Oplophoroidea, but the larval development is unknown for most of them. The complete larval development of *Systellaspis debilis* (*Milne-Edwards, 1881*), here described and illustrated, is the first one to have been successfully reared in the laboratory, consisting of four zoeal and one decapodid stages. The zoeae were found to be fully lecithotrophic, which together with the females' lower fecundity, are probably evolutionary consequences of the species mesopelagic habitat.

## INTRODUCTION

*Systellaspis debilis* is a mesopelagic shrimp belonging to the family Oplophoridae, occurring in the Atlantic and Indian Oceans (*Lunina, Kulagin & Vereshchaka, 2018*). Even though a lot is known about the mesopelagic community and its function in marine ecosystems, much remains unknown, especially for the crustacean species. However, oplophorid shrimps and their larvae gained attention recently with the application of molecular techniques to the phylogenetic systematic studies of Caridea. *Wong et al. (2015)* provided the molecular data to support the resurrection of the family Acanthephyridae by *Chan et al. (2010)*, retrieving two major clades within the Oplophoroidea, the Oplophoridae and the Acanthephyridae. Recently, *Lunina, Kulagin & Vereshchaka (2018)* presented the most comprehensive phylogenetic analyses for this group to date, using morphological and molecular data, showing four robustly supported species groups within the genus *Systellaspis*.

Seventy years earlier, *Gurney & Lebour (1941)* suggested to separating the oplophorid species into two groups according to the size of the eggs and the number of zoeal stages in the life cycle: the first group including the genera *Oplophorus*, *Systellapis*, *Ephyrina* and *Hymenodora* having large lipid-filled eggs and five or fewer zoeal stages, and the second group, with the genera *Acanthephyra*, *Meningodora* and *Notostomus* having small eggs and nine or more planktotrophic stages. Previously, *Kemp (1907)* noticed that *Systellaspis debilis*

Corresponding author
Cátia Bartilotti, cbartilotti@ipma.pt

had much larger eggs and their newly hatched larvae were more developed than those of *Acantephyra purpurea*. The egg size is an important aspect of the life history of marine organisms, with large eggs generally reflecting an increased maternal investment (e.g., *Moran & McAlister, 2009*).

Of the 16 valid species of Oplophoridae Dana, 1852, only two larval sequences collected from plankton have been described: five zoeal stages and one decapodid for *Oplophorus spinosus*, and four zoeal stages and one decapodid for *Systellaspis debilis*, both by *Gurney & Lebour (1941)*. *Systellaspis debilis* first zoeal stage is described as having a small rostrum, eyes large and sessile, all pereopods and pleopods present as buds, a broad telson with a small median indent with 7 + 7 spines fringed with spinules, and the uropods visible under integument (*Gurney & Lebour, 1941*). The second stage is presented as having eyes stalked, the peduncle of the antennule 3-segmented, the telson with 8 + 8 spines and free uropods. Regarding the third larval stage, *Gurney & Lebour (1941)* described it as having the rostrum shorter than the antennal scaphocerite with 10 small dorsal teeth, the flagellum of the antennule long, the first and second pereopods chelate and the uropods reaching the convex distal margin of a parallel-sided telson. No intermediate stage was described between the second and the third zoea, but it was registered that the second zoea corresponds to a normal third zoeal stage of a caridean larvae (*Gurney & Lebour, 1941*). Finally, some figures of a possible fourth zoeal stage and the description of the "Post-larval 1," which corresponds to nowadays decapodid stage, with the thorax still full of yolk, are presented (*Gurney & Lebour, 1941*).

Gathering information on the biological traits for mesopelagic species is difficult, since their capture and live maintenance under laboratory conditions are problematic. However, larval development under laboratory conditions for morphological purposes can be a useful way to acquire knowledge on feeding traits and development patterns. Throughout the adult phase of the life-cycle, during the oogenesis the energy for partially or entirely food-independent larval development is stored as egg yolk, establishing the degree of lecithotrophy or planktotrophy (e.g., *Anger, 2001*). Therefore, full lecithotrophy and planktotrophy are extremes in a continuum: full lecithotrophy corresponds to high yolk stores that allow no feeding during the development of the larvae, and planktotrophy corresponds to larvae that need to feed during their development (e.g., *Anger, 2001*).

The present work aims to describe the complete larval development of *Systellaspis debilis* from laboratory reared material, comparing it with the previous description available for the species, and presenting information on the development pattern of the larvae. Also, we discuss the implications of the lecithotrophy and planktotrophy of the larvae on the ecology and evolution of the oplophorid shrimps.

## MATERIALS AND METHODS

### Specimen collection and larval culture

The ovigerous females of *Systellaspis debilis* were collected in two different research surveys carried out by Instituto Português do Mar e da Atmosfera, I.P. (IPMA, former IPIMAR) off the southwest and southern coasts of Portugal, in August 2010 and June 2011 onboard of RV Noruega, as a by-catch of the plankton sampling (Bongo net, with 90 cm of

**Table 1 Sampled females used in present study.** Sampled females used in present study, per date of collection; number of eggs; stage of the eggs—defined as early stage (1), with intense color and no other pigmentation visible, middle stage (2), with a more pale color and an embryo having a slight eye pigmentation and late stage (3), almost without color and with an embryo with the eye pigmentation well visible and developed (adapted from *Company & Sardà (1997)*); size (length, corresponding to the major axis distance × width, corresponding to the minor axis distance, in mm) and volume of the eggs; number of hatchings; and date of hatching.

| Female | Date of collection | Number of eggs | Stage of the eggs | Size of the eggs (in mm) | Volume of the eggs (in mm$^{-3}$) | Number of hatchings | Date of hatching |
|---|---|---|---|---|---|---|---|
| 1 | August 15, 2010 | 12 | 1 | 3.4–3.6 × 1.9–2.1 | 6.43–8.31 | 0 | – |
| 2 | August 15, 2010 | 8 | 3 | 3.6 × 2.1 | 8.31 | 5 | August 20, 2010 |
| 3 | August 15, 2010 | 9 | 3 | 3.5–3.6 × 2.1 | 8.08–8.31 | 2 | September 7–8, 2010 |
| 4 | August 15, 2010 | 7 | 1 | 3.6–3.8 × 2.0 | 7.54–7.96 | 0 | – |
| 5 | August 15, 2010 | 10 | 2 | 3.5 × 2.0 | 7.33 | 4 | September 21, 2010 |
| 6 | June 28, 2011 | 9 | 3 | 3.7 × 2.0 | 7.75 | 8 | July 1, 2011 |
| 7 | June 28, 2011 | 7 | 2 | 3.5–3.9 × 2.0 | 7.33–8.17 | 6 | July 25, 2011 |
| 8 | June 28, 2011 | 3 | 2 | 3.6–3.9 × 2.0 | 7.54–8.17 | 0 | – |
| 9 | June 28, 2011 | 2 | 2 | 3.5–3.9 × 2.1 | 8.08–9.01 | 0 | – |
| 10 | June 29, 2011 | 11 | 2 | 3.6 × 2.1 | 8.31 | 1 | July 29, 2011 |
| 11 | June 21, 2011 | 12 | 3 | 3.7 × 2.0 | 7.75 | 9 | July 4–6, 2011 |
| 12 | June 28, 2011 | 10 | 2 | 3.5–4.0 × 2.0 | 7.33–8.38 | 8 | July 25, 2011 |
| 13 | June 21, 2011 | 12 | 3 | 3.6–3.9 × 2.0 | 7.54–8.17 | 10 | July 8–9, 2011 |
| | | Total: 112 | – | Average: 3.65 ± 0.09 × 2.0 ± 0.05 | Average: 7.90 ± 0.53 | Total: 53 | – |

diameter aperture, and 500 μm of mesh size) and the crustacean bottom trawls (field permit approval by IPMA, Oceanographic Survey MedEx-02060810, and Nephrops Survey Offshore Portugal-02030611, respectively). A total of 13 ovigerous females were obtained in both years. All females were captured dead, each carrying between a minimum of two and a maximum of 12 eggs (Table 1) that were immediately and carefully removed from the pleopods and transferred to 200 ml glass beakers with aerated autoclaved seawater. Following *Company & Sardà (1997)*, three stages of egg development were defined: the early stage (1), with intense color and no other pigmentation visible; the middle stage (2), an egg with a more pale color and with an embryo having a slight eye pigmentation; and the late stage (3), an egg almost without color and with an embryo with the eye pigmentation well visible and developed. In stages 2 and 3, the heartbeat was checked in order to know if the embryo was alive. Once at the laboratory, the elliptical eggs were measured on the major and the minor axes, corresponding, respectively, to its length and width, under a binocular microscope with a micrometer lens. Once at the laboratory, the elliptical eggs were measured on the major and the minor axes, corresponding, respectively, to its length and width, under a binocular microscope with a micrometer lens. The eggs were kept in autoclaved seawater with a salinity of 35 ± 1, a temperature of 18 ± 1 °C, a photoperiod of 12 h light:12 h dark, and weak aeration. The eggs were checked daily until hatching. Once hatched, the larvae were separated with the aid of a glass pipette and transferred to a 30 ml glass beaker, to be individually reared with constant weak aeration. During rearing, the larvae were kept in exactly the same conditions. The water was changed daily, and the larvae checked for the evidence of molting (presence of

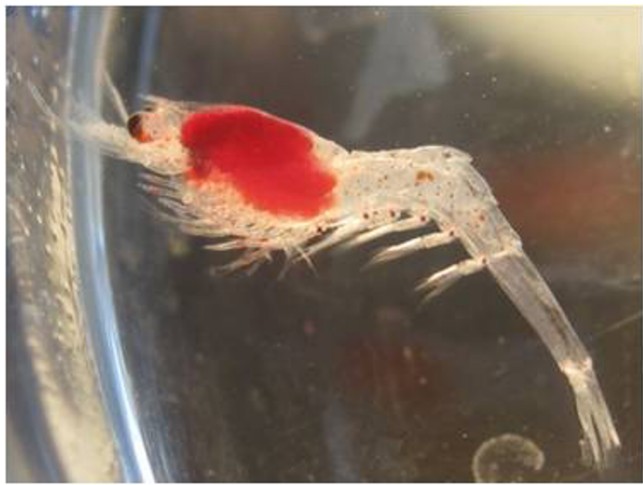

**Figure 1** *Systellaspis debilis*, **decapodid.** Yolk stored in the pereon.

exuviae in the bottom of the culture recipient). In experiments carried out in 2010, a high hatching efficiency *Artemia* sp. (250,000 nauplii.g$^{-1}$ of product) prepared following *Sorgeloos, Dhert & Candreva (2001)*, were provided daily at a density of 10 nauplii.ml$^{-1}$. No decrease in the density of *Artemia* was verified, leading us to suppose that these were nonfeeding larvae. Since it was observed that the larval cycle was completed, the experience was repeated in 2011 without food. In both years, the experiments finished with the larvae reaching the decapodid and juvenile stages, which were fixed in 4% borax buffered formaldehyde along with the exuviae for later morphological analysis.

### Larval drawings and measurements

Drawings and measurements were made following the method described in detail by *Bartilotti, Salabert & Dos Santos (2016)*. The long plumose setae on the exopods of maxillipeds, and on the pleopods and uropods were drawn truncated, and the setules from setae were omitted from drawings when necessary. The number of examined specimens per stage (N) is referred in the description. Measurements taken were: total length (TL), corresponding to the distance from the tip of the rostrum to the posterior end of telson; carapace length (CL), measured from the tip of rostrum to the posterior margin of the carapace; and rostrum length (RL), corresponding to the distance from the tip of rostrum to the eye socket (except in zoea I where it was not measured). Ten specimens of first to third zoeal stages, seven specimens of fourth zoeal stage, and six specimens of decapodid stage were measured. The larval series has been deposited in IPMA-Instituto Português do Mar e da Atmosfera, in Lisbon, Portugal (IPIMAR/O/Sd/01/2011).

## RESULTS

### Rearing the larvae of *Systellaspis debilis*

Under laboratory conditions the larvae hatched with a high amount of bright red yolk, stored in the pereon and observed throughout all the development (Fig. 1). Hatching

occurred 3–40 days after the collection of the eggs (Table 1). During the larval rearing, both zoea and decapodid, stood still, floating most of the time and swimming only as a response to an external stimulus (e.g., tactile stimuli with an autoclaved glass pipette).

In 2010 rearing experiments, no feeding behavior was observed, and the larvae went through the entire larval cycle using their yolk. Therefore, no food was provided in the 2011 rearing experiments, but the decapodid stage was successfully achieved, after the four zoeal stages. Thus, we conclude that this species has a full lecithotrophic development. Considering the 2 years of experiments, with an average duration of 28.23 ± 1.48 days after hatching, the zoeae were able to molt and grow, going through the metamorphosis to the decapodid. The duration of the successive stages was 4–6 days for stage I, 5–8 days for stages II and III, 8–11 days for stage IV. After 12 days, two of the decapodid stage larvae were able to molt, one in 2010 and another in 2011, reaching the first juvenile stage without food.

## Description of the complete larval development of *Systellaspis debilis*

Under laboratory conditions at 18 ± 1 °C of temperature, four zoeal stages and one decapodid were identified, and are described in detail.

### *Systellaspis debilis* (Milne-Edwards, 1881)
(Figs. 2–7)

#### First zoea

Dimensions: TL = 9.35–10.01 mm; CL = 3.19–3.52 mm; $N$ = 5.

Carapace (Figs. 2A–2C): rostrum small, down turned; eyes compound and sessile; pterigostomian spine present, followed by 8–9 very small denticles.

Antennule (Fig. 2D): peduncle unsegmented, with 1 terminal plumose seta and short outer flagellum with 1 plumose seta and 2 shorter + 1 longer aesthetascs terminally.

Antenna (Figs. 2E and 2F): protopod unsegmented, with a small strong distal papposerrate seta; endopod less than half the length of the scaphocerite, apically with 1 plumose seta; scaphocerite unsegmented, broad, with 24–26 plumose setae and 1 small spine on apex.

Mandibles (Fig. 2G): slightly asymmetrical, with undifferentiated molar and incisive processes; palp present.

Maxillule (Fig. 2H): coxal endite with 1–2 small spines, basial endite with 1 small spine; endopod unsegmented with 1 short simple seta at half the length, 1 spine subterminally and 2 plumose setae terminally.

Maxilla (Fig. 2I): coxal endite bilobed with 2 simple + 2–3 plumose setae, basial endite bilobed with 1 simple and 1 plumose + 2 simple and 1 plumose setae; endopod unsegmented bearing 2 + 1 + 1 + 3 plumose setae; scaphognathite with 8–10 marginal plumose setae, and one long distal stout setose process.

First maxilliped (Fig. 2J): coxa with 2–3 simple setae; basis with 7–8 simple setae; endopod 4-segmented with 1, 1, 1 + 1, 1 + 3 plumose setae; exopod unsegmented,

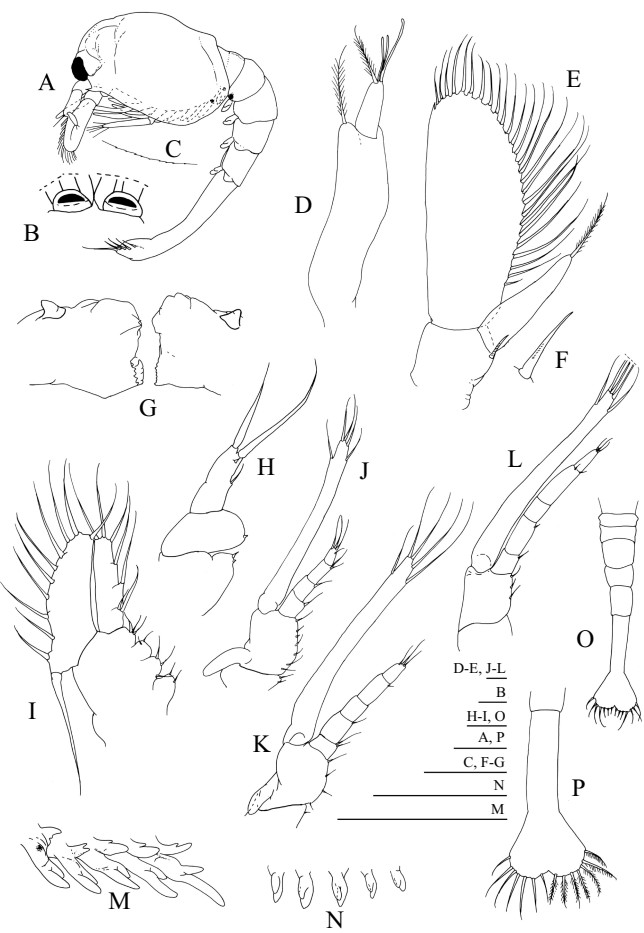

**Figure 2 *Systellaspis debilis*, first zoea.** (A) Total animal, lateral view, (B) detail of rostrum, dorsal view, (C) detail of carapace anterior ventral margin, (D) antennule, (E) antenna, (F) detail of antennal protopod seta, (G) mandibles, (H) maxillule, (I) maxilla, (J) first maxilliped, (K) second maxilliped, (L) third maxilliped, (M) first to fifth pereopods, (N) pleopods, (O) pleon with telson, (P) telson. Scale bars: 0.05 mm (F–G); 0.1 mm (A–E, H–P).

bearing 2 shorter plumose setae subterminally and 3 plumose setae terminally; epipod present.

Second maxilliped (Fig. 2K): coxa with 1 simple seta; basis with 3–4 simple setae; endopod 5-segmented with 2, 1, 0, 1 and 3 simple terminal setae; exopod unsegmented, bearing 2 subterminal and 3 terminal plumose setae; epipod present.

Third maxilliped (Fig. 2L): coxa unarmed; basis with 2–3 simple setae; endopod 5-segmented with 1–2, 1, 0, 1–2 and 3 simple setae; exopod unsegmented, bearing 2 subterminal and 3 terminal plumose setae.

First to fifth pereopods (Fig. 2M): biramous buds, with pleurobranchs present; photophores present on the fifth pair of pereopods.

Pleon (Figs. 2A and 2O): 5 pleomeres, without setae or spines, fifth pleomere with a rounded pleura.

Pleopods (Figs. 2A and 2N): present as biramous buds; photophores present on first and second pleopods.

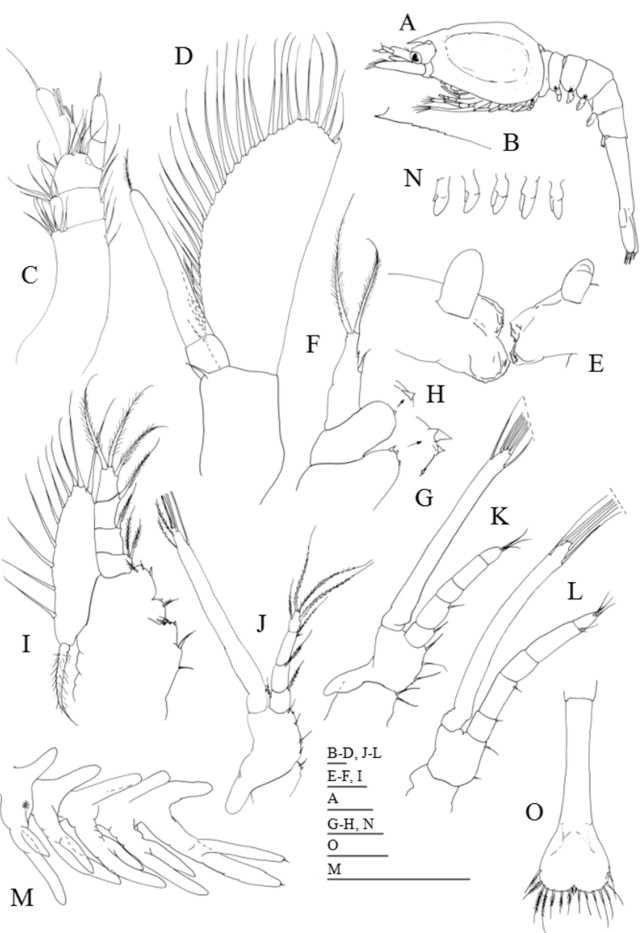

**Figure 3** *Systellaspis debilis,* **second zoea.** (A) Total animal, lateral view, (B) detail of carapace anterior ventral margin, (C) antennule, (D) antenna, (E) mandibles, (F) maxillule, (G) detail of maxillule coxal endite, (H) detail of maxillule basial endite, (I) maxilla, (J) first maxilliped, (K) second maxilliped, (L) third maxilliped, (M) first to fifth pereopods, (N) pleopods, (O) telson. Scale bars: 0.05 mm (G–H); 0.1 mm (A–F, I–L, O); 1 mm (M–N).

Uropods: absent.

Telson (Figs. 2O and 2P): triangular, broader posteriorly, with a small median indent, with 7 + 7 setae, the inner 5 plumose and the outer 2 plumose on proximal axis only.

### Second zoea

Dimensions: TL = 10.34–11.00 mm; CL = 3.52–3.85 mm; RL = 0.66–0.88; $N$ = 5.

Carapace (Figs. 3A and 3B): eyes stalked, with the ocular peduncle reaching half the length of the antennal peduncle; rostrum triangular, as long as the eyes; pterigostomian spine followed by 8–9 small denticles.

Antennule (Fig. 3C): peduncle 3-segmented, with 1 small spine positioned at about two-thirds of the length of the first segment + 3 plumose setae along the inner margin + 7–8 plumose setae distributed on distal outer margin, second segment with 2 plumose setae along the inner margin + 5–6 plumose setae distributed on distal outer margin, and distal segment with 1 plumose seta on inner margin + 1 short plumose and 1 simple setae

on outer margin + 6–7 plumose setae distally; inner flagellum 2-segmented, first segment naked and distal segment bearing 1 simple seta distally; outer flagellum unsegmented with 2 aesthetascs at one-third of the length of the segment + 2 aesthetascs at two-thirds of the length of the segment + 1 simple seta terminally.

Antenna (Fig. 3D): protopod unsegmented with a small strong simple spine; 2-segmented endopod longer than half the length of the scaphocerite, shorter basal segment naked, terminal longer segment with 1 plumose seta distally; scaphocerite with 30–31 plumose setae and 1 strong spine on apex.

Mandibles (Fig. 3E): palp enlarged in size, otherwise unchanged.

Maxillule (Figs. 3F–3H): coxal endite with 1–3 spines, basial endite with 1 spine; endopod unchanged.

Maxilla (Fig. 3I): coxal endite bilobed with 2–3 simple + 1 spine and 2–3 plumose setae distributed as illustrated; basial endite bilobed with 1 plumose seta and 1 + 1 spines, 1 simple and 1 plumose setae; endopod unsegmented bearing 2 + 1 + 1 + 3 plumose setae; scaphognathite with 10–13 marginal plumose setae, and one long distal stout setose process.

First maxilliped (Fig. 3J): coxa with 1 small spine and 1 plumose seta; basis with 2 small spines and 4–6 plumose setae distributed as illustrated; endopod 4-segmented with 1 plumose seta on inner margin and 1 plumose seta on outer margin, 1 plumose seta, 1 simple and 1 plumose setae and 1 + 3 plumose setae; exopod unsegmented, bearing 2 shorter setae subterminally and 4 long terminal plumose setae; epipod unchanged.

Second maxilliped (Fig. 3K): coxa with 1–2 plumose setae; basis with 3–4 plumose setae; endopod 5-segmented with 1, 1, 0, 0–1, and 1 + 3 plumose setae; exopod unsegmented bearing 2 shorter setae subterminally and 4 long terminal plumose setae; epipod unchanged.

Third maxilliped (Fig. 3L): coxa unarmed; basis with 2–3 plumose setae; endopod 5-segmented with 1, 1, 0, 2 and 3 plumose setae; exopod unsegmented, bearing 2 subterminal and 4 terminal plumose setae.

First to fifth pereopods (Fig. 3M): first pereopod endopod and exopod bearing one small simple seta apically, otherwise unchanged besides size.

Pleon (Fig. 3A): unchanged besides size.

Pleopods (Figs. 3A and 3N): biramous buds enlarged in size; photophores present on first to third pleopods.

Uropods: absent, but already visible under telson integument.

Telson (Fig. 3O): with a small median indentation, now with 8 + 8 setae, the inner 7 plumose and the outer plumose on proximal axis only.

### Third zoea

Dimensions: TL = 10.45–11.00 mm; CL = 3.63–3.96 mm; RL = 0.77–0.88; $N$ = 6.

Carapace (Figs. 4A and 4B): pterigostomian spine followed by 8–9 small denticles on lateral antero-ventral margin.

Antennule (Fig. 4C): peduncle 3-segmented, basal segment with 1 spine positioned at half the length of the segment, 1–2 plumose setae on inner margin, 1 small plumose seta

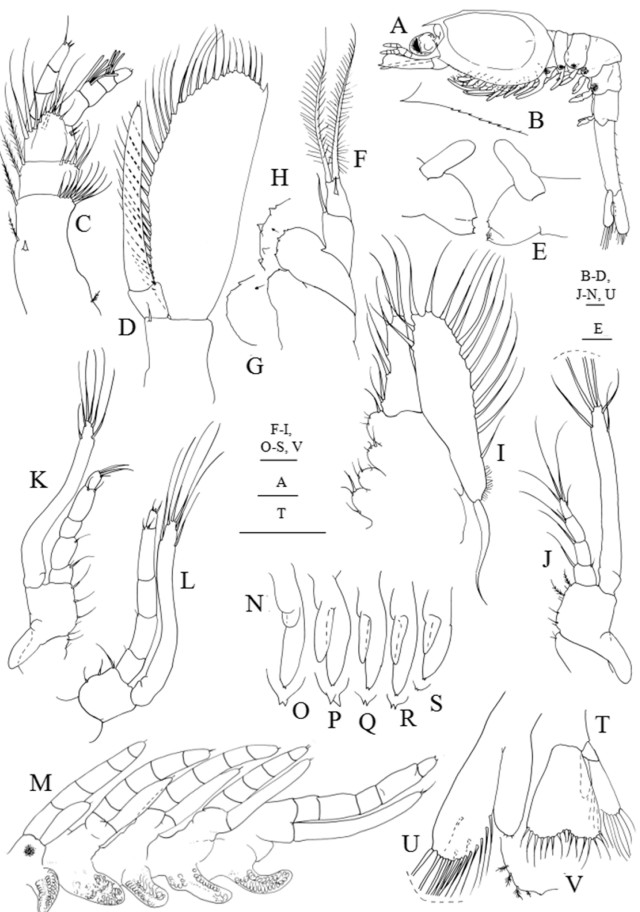

**Figure 4 Systellaspis debilis, third zoea.** (A) Total animal, lateral view, (B) detail of carapace anterior ventral margin, (C) antennule, (D) antenna, (E) mandibles, (F) maxillule, (G) detail of maxillule coxal endite, (H) detail of maxillule basial endite, (I) maxilla, (J) first maxilliped, (K) second maxilliped, (L) third maxilliped, (M) first to fifth pereopods, (N) pleopods, (O) detail of first pleopod exopod posterior margin, (P) detail of second pleopod exopod posterior margin, (Q) detail of third pleopod exopod posterior margin, (R) detail of fourth pleopod exopod posterior margin, (S) detail of fifth pleopod exopod posterior margin, (T) telson and uropods, (U) detail of uropods, (V) detail of uropods endopod posterior margin. Scale bars: 0.05 mm (G–H, O–S, V); 0.1 mm (A–F, I–N, T–U).

on the stylocerite and 9–12 plumose setae distributed on distal outer margin; second segment with 2 plumose setae along the inner margin and 5–7 plumose setae distributed on distal outer margin; distal segment with 2 + 2 plumose setae along inner margin, 6 short plumose and 4 plumose setae on distal outer margin; inner flagellum 4-segmented, first and second segments naked, distal segment bearing 2 simple subterminal + 1 plumose terminal setae; outer flagellum 4-segmented with 1, 3 and 3 aesthetascs in the first three segments, and 1 + 1 simple setae terminally.

Antenna (Fig. 4D): protopod unsegmented with a strong spine; 2-segmented endopod longer than three quarters of the length of the scaphocerite, shorter basal segment naked, terminal longer segment with 2–3 very small simple setae terminally; scaphocerite with 32–34 plumose setae and 1 strong spine on apex.

Mandibles (Fig. 4E): palp enlarged in size, otherwise unchanged.

Maxillule (Figs. 4F–4H): coxal endite with 3 very small spines, basial endite with 4–5 spines; 2-segmented endopod, proximal segment with 1 simple seta, distal segment with 1 spine subterminally and 2 plumose setae terminally.

Maxilla (Fig. 4I): coxal endite bilobed with 3–4 + 3 simple and plumose setae distributed as illustrated, basial endite bilobed with 2–3 + 3–4 simple and plumose setae; endopod unsegmented bearing 2 + 1 + 1 + 3 plumose setae; scaphognathite with 15–17 marginal plumose setae, microtricha as illustrated, and one long distal stout setose process.

First maxilliped (Fig. 4J): coxa with 2 simple setae; basis with 7–8 simple and plumose setae distributed as illustrated; endopod 4-segmented with 1 + 1 plumose setae on inner margin and 1 plumose seta on outer margin, 1 plumose seta, 1 simple and 1 plumose setae and 1 shorter subterminal + 3 longer terminal plumose setae; exopod unsegmented, bearing 2 shorter setae subapically and 4 long terminal plumose setae; epipod enlarged in size.

Second maxilliped (Fig. 4K): coxa with 1–2 plumose setae; basis with 4–5 plumose setae; endopod 5-segmented with 2, 1, 0–1, 1 + 1 and 1 + 2 plumose setae; exopod unsegmented bearing 2 shorter setae subapically and 4 long terminal plumose setae; epipod enlarged in size.

Third maxilliped (Fig. 4L): coxa unarmed; basis with 3–4 plumose setae; endopod 5-segmented with 1–2, 1, 0, 1 + 1 and 3 plumose setae; exopod unsegmented, bearing 2 subterminal and 4 terminal plumose setae.

First to fifth pereopods (Fig. 4M): first pereopod endopod 5-segmented, subchelate, with internal distal margin of propodus produced forward to about one-third of dactylus, with 1 + 1 simple setae distally on terminal segment, and exopod unsegmented bearing 2 simple setae apically; second to fifth pereopods endopod 5-segmeted bearing 1–2 very small simple setae terminally, and exopod unsegmented with 1–2 very small simple setae terminally; photophores present on the fifth pair of pereopods.

Pleon (Fig. 4A): first to fifth pleomeres unchanged; sixth pleomere separated from the telson with six pairs of small simple setae distributed as figured.

Pleopods (Figs. 4A and 4N–4S): enlarged in size; first pleopod endopod rudimentary, bud-like and exopod with 1 small spine subterminally and 2 small spines terminally; second to fifth pleopods endopod rudimentary, bud-like and exopod with 2 small spines terminally. Photophores present on first to fourth pleopods.

Uropods (Figs. 4A and 4T–4V): biramous; endopod small with 2–3 short plumose seate apically; exopod well developed shorter than the posterior margin of telson, with 12–14 marginal plumose setae and 2–3 plumose setae on ventral margin.

Telson (Figs. 4A and 4T): separated from the sixth pleomere, with 8 + 8 setae.

### Fourth zoea

Dimensions: TL = 10.80–11.28 mm; CL = 3.84–4.00 mm; RL = 0.80–1.04; $N$ = 4.

Carapace (Figs. 5A and 5B): three luminous organs on eye; rostrum extending beyond the eyes, with 9–10 dorsal teeth and 2 very small ventral teeth; pterigostomian spine followed by 7–9 very small spines on lateral ventral margin.

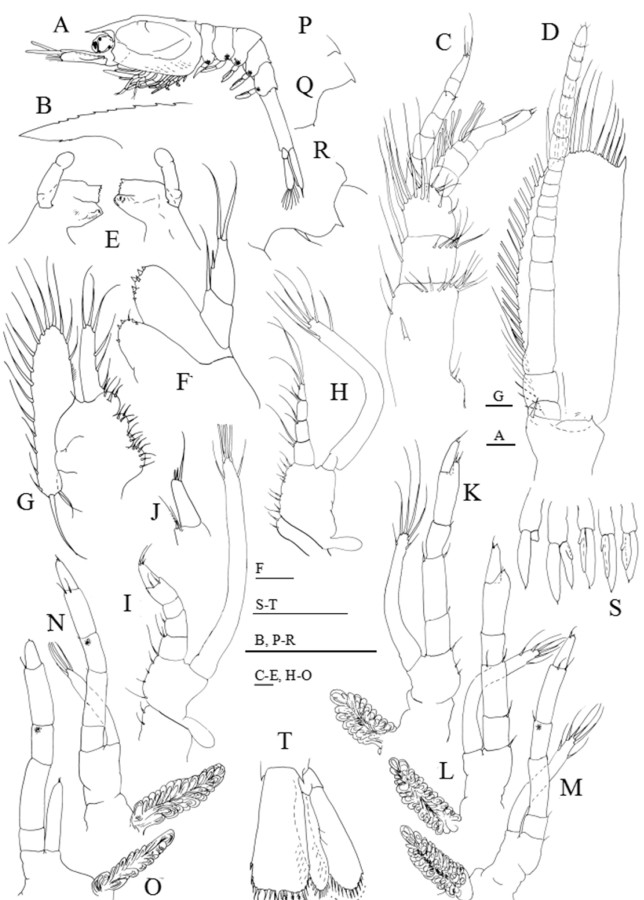

**Figure 5 *Systellaspis debilis*, fourth zoea.** (A) Total animal, lateral view, (B) detail of rostrum, (C) antennule, (D) antenna, (E) mandibles, (F) maxillule, (G) maxilla, (H) first maxilliped, (I) second maxilliped, (J) detail of propodus posterior end and dactylus of third maxilliped, (K) first pereopod, (L) second pereopod, (M) third pereopod, (N) fourth pereopod, (O) fifth pereopod, (P) detail of fourth pleomere posteromedial margin, (Q) detail of fifth pleomere posterior margin, (R) detail of sixth pleomere posterior margin, (S) pleopods, (T) telson and uropods. Scale bars: 0.1 mm (A, C–O); 1 mm (B, P–R, S–T).

Antennule (Fig. 5C): peduncle 3-segmented, basal segment with 1 spine positioned at two-thirds of the length of the segment, 2–3 plumose setae on inner margin, 2 small plumose setae on the stylocerite, and 11–13 plumose setae distributed on distal outer margin; second segment with 2–3 plumose setae along the inner margin and 6–7 plumose setae distributed on distal outer margin; distal segment with 8 subterminal plumose + 6 terminal plumose setae distributed as illustrated; inner flagellum 6-segmented, first to third segments naked, third and fourth segments with 2–3 simple setae, and terminal segment with 3–4 simple setae; outer flagellum 5-segmented, respectively, with 1 aesthetasc, 2 + 3 aesthetascs, 3 aesthetascs, 1 simple seta and 2–3 simple setae.

Antenna (Fig. 5D): protopod unsegmented with one strong simple seta on the inner margin and one strong spine on the outer margin; endopod measuring almost one and a half times the length of the scaphocerite, with 18 segments each bearing 0–4 very small

simple setae terminally; scaphocerite now with 35–36 plumose setae and 1 strong spine on apex.

Mandibles (Fig. 5E): incisor and molar processes as illustrated; palp enlarged in size, unsegmented.

Maxillule (Fig. 5F): coxal endite with 6–7 spines, basial endite with 7–8 spines; 2-segmented endopod, proximal segment with 1 simple seta, distal segment with 1 spine subterminally and 2 plumose setae terminally.

Maxilla (Fig. 5G): coxal endite bilobed with 5–6 + 3 simple and plumose setae distributed as illustrated; basial endite bilobed with 3–4 + 3–4 simple and plumose setae; endopod unsegmented bearing 2 + 1 + 1 + 3 plumose setae; scaphognathite with 17–20 marginal plumose setae, and two distal stout setose processes.

First maxilliped (Fig. 5H): coxa with 2 simple setae; basis with 8–9 simple and plumose setae distributed as illustrated; endopod 4-segmented with 1 + 1 plumose setae, 1 + 1 plumose setae, 1 + 1 plumose setae and 1 shorter subterminal + 3 longer terminal plumose setae; exopod unsegmented, bearing 2 shorter setae subapically and 4 long terminal plumose setae; epipod enlarged in size.

Second maxilliped (Fig. 5I): coxa with 1–2 plumose setae; basis with 5–6 plumose setae; endopod 5-segmented with 2, 1, 0–1, 2 and 1 + 2 plumose setae; exopod unsegmented bearing 2 shorter setae subapically and 4 long terminal plumose setae; epipod enlarged in size.

Third maxilliped (Fig. 5J): coxa unarmed; basis with 3–4 plumose setae; endopod 5-segmented with 1–2, 1, 0, 2 and 3 plumose setae, and a luminous organ on the fourth segment (propodus); exopod unsegmented, bearing 2 subterminal and 4 terminal plumose setae.

First pereopod (Fig. 5K): functional, coxa unarmed; basis with 3 simple setae; endopod subchelate, 5-segmented, ischium, merus and carpus with 1, 1–2, 1 simple setae, respectively, as illustrated; distal margin of the propodus produced forward to about one-third of dactylus bearing 1–2 simple setae + 1 spinous process, dactylus with 1 simple seta + 1 spinous process; exopod unsegmented, bearing 2 subterminal and 4 terminal plumose setae; pleurobranch developed.

Second pereopod (Fig. 5L): functional, coxa unarmed; basis with 1 simple seta; endopod subchelate, 5-segmented, ischium, merus and carpus with 1, 1–2 and 0 simple setae, respectively, distal margin of the propodus produced forward to about one-third of dactylus bearing 1 simple seta + 1 spinous process, dactylus with 1 simple seta + 1 spinous process; exopod unsegmented, bearing 2 subterminal and 4 terminal plumose setae; pleurobranch developed.

Third pereopod (Fig. 5M): functional, coxa unarmed; basis with 1 simple seta; endopod 5-segmented, ischium, merus, carpus and propodus with 1, 0–1, 1, 2 simple setae, dactylus with 1 simple seta + 1 spinous process; a luminous organ on the carpus; exopod unsegmented, bearing 2 subterminal and 4 terminal plumose setae; pleurobranch developed.

Fourth pereopod (Fig. 5N): functional, coxa unarmed; basis with 1 simple seta; endopod 5-segmented, ischium, merus, carpus and propodus with 0–1, 1, 1, 2 simple setae, dactylus

with 1 simple seta + 1 spinous process; a luminous organ on the carpus; exopod unsegmented, bearing 4 terminal plumose setae; pleurobranch developed.

Fifth pereopod (Fig. 5O): functional, coxa unarmed; basis naked; endopod 5-segmented, ischium, merus, carpus and propodus with 0, 0, 0, 1 simple seta, dactylus ending in a spinous process; a luminous organ on the carpus; exopod unsegmented, bearing 2 small terminal simple setae; pleurobranch developed.

Pleon (Figs. 5A and 5P–5R): first to third pleomeres unchanged; fourth pleomere with a posteromedial spine; fifth pleomere with a posteromedial spine and a spine on the pleura; sixth pleomere with a pair of lateral spines.

Pleopods (Figs. 5A and 5S): protopod naked; first pleopod endopod naked and exopod with 2 small spines; second to fifth pleopods endopods and exopods with 2 small spines distally, *appendix interna* as bud, present from second to fifth pleopods. Photophores present on first to fifth pleopods.

Uropods (Fig. 5T): protopod without setae; endopod with 2–3 short plumose setae medially on outer margin and 14–16 plumose setae along distal and inner margins; exopod, with 1 small spine on apex followed by 20–22 plumose setae along distal and inner margin, and 2–3 plumose setae on ventral margin.

Telson (Fig. 5T): with a median spine on posterior margin, followed by five pairs of strong plumose processes on the posterior end being the fourth pair plumose only on the inner margin, two pairs of outer spines and one pair of lateral spines; almost rectangular shaped.

### Decapodid

Dimensions: TL = 11.77–12.10 mm; CL = 4.40–4.62 mm; RL = 1.10–1.32; N = 4.

Carapace (Figs. 1 and 6A–6C): eyestalk with ocellus on medio-distal portion, and three luminous organs; rostrum shorter than scaphocerite with 9–10 dorsal teeth and 2–3 small ventral teeth; antennal spine present, and pterigostomian spine followed by 10–11 small spines on lateral ventral margin.

Antennule (Fig. 6H): peduncle 3-segmented, basal segment with 1 spine positioned at two-thirds of the length of the segment, 3–5 plumose setae on inner margin, 1–2 small plumose setae on the stylocerite and 11–13 plumose setae distributed on distal outer margin; second segment with 2–3 plumose setae along the inner margin and 6–8 plumose setae distributed on distal outer margin; distal segment with 7–9 subterminal plumose + 7–8 terminal plumose setae distributed as illustrated; inner flagellum with 10–11 segments each with 0–5 simple setae; outer flagellum with 10–11 segments, first three segments with 1 aesthetasc, 2 + 3 aesthetascs + 1 simple seta and 3 + 3 aesthetascs, fourth to last segment naked to 3 simple setae distally.

Antenna (Fig. 6I): protopod 2-segmented, distal segment with one simple seta and one strong spine; endopod longer than the scaphocerite, with 33–36 segments each bearing 0–5 very small simple setae terminally; scaphocerite now with 39–41 plumose setae, 1 strong spine on apex, and 1–2 simple setae distributed along outer margin.

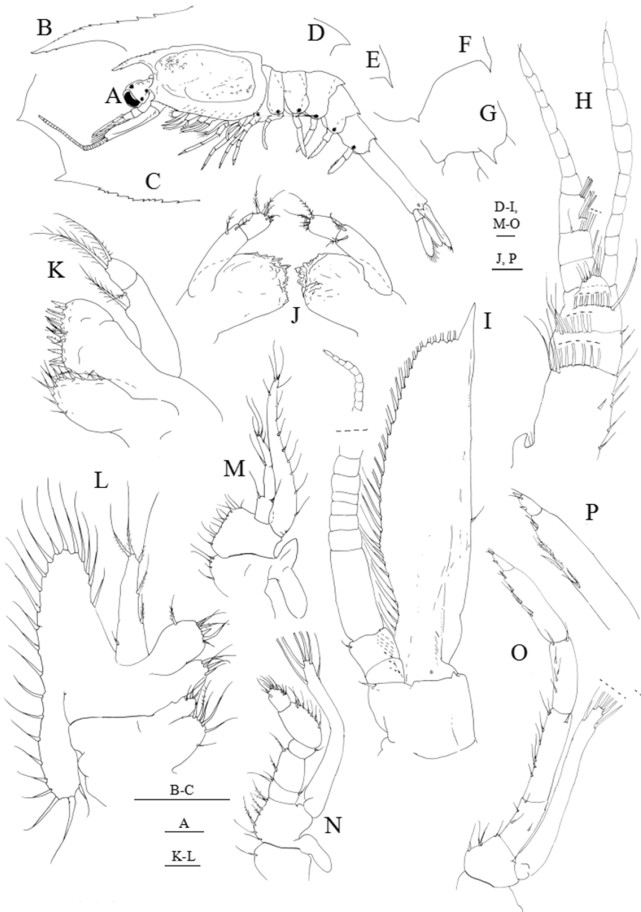

**Figure 6 Systellaspis debilis, decapodid.** (A) Total animal, lateral view, (B) detail of rostrum, (C) detail of carapace anterior ventral margin, (D) detail of third pleomere posteromedial margin, (E) detail of fourth pleomere posteromedial margin, (F) detail of fifth pleomere posterior margin, (G) detail of sixth pleomere posterior margin, (H) antennule, (I) antenna, (J) mandibles, (K) maxillule, (L) maxilla, (M) first maxilliped, (N) second maxilliped, (O) third maxilliped, (P) detail of propodus and dactylus of third maxilliped. Scale bars: 0.1 mm (A, D–P); 1 mm (B–C).

Mandibles (Fig. 6J): palp 2-segmented, proximal segment with 2 plumose setae, distal segment with 2 simple + 4 plumose setae; incisor process with about 10–12 strong serrated teeth, molar process as illustrated.

Maxillule (Fig. 6K): coxa with 10–12 simple and papposerrate setae; basis with 15–17 cuspidate and papposerrate setae; 2-segmented endopod, proximal segment with 1 longer subterminal and 1 shorter terminal outer plumose setae, distal segment with 2 plumose setae terminally.

Maxilla (Fig. 6L): coxal endite bilobed with 6–7 + 3 setae; basial endite bilobed with 5–6 + 6–7 setae; endopod unsegmented bearing 1 + 1 + 1 and 3 plumose setae terminally; scaphognathite with 21–23 marginal plumose setae.

First maxilliped (Fig. 6M): coxa with 3–4 setae; basis with 12–13 papposerrate setae; endopod 3-segmented, with 1 + 1, 1, 1 + 3 papposerrate setae distributed as figured;

exopod with 9–10 short plumose setae on outer margin, 3 terminal plumose setae and 1 shorter plumose seta subapically on inner margin; epipod bilobed.

Second maxilliped (Fig. 6N): coxa with 1 plumose seta; basis with 5–7 plumose setae; endopod 5-segmented with 1–2 + 1, 1 + 1, 0–1, 6–7 and 10–11 plumose setae; exopod unsegmented bearing 2 setae subapically and 4 long terminal plumose setae; epipod enlarged in size.

Third maxilliped (Figs. 6O and 6P): coxa unarmed; basis with 3–4 plumose setae; endopod 5-segmented with 4–5 plumose setae, 1 spine + 6–7 plumose setae, 5–6 plumose setae, 8–10 plumose setae and 2 spines + 2 simple setae distributed as illustrated, and a luminous organ on the fourth segment (propodus); exopod unsegmented, bearing 2 subterminal and 4 terminal plumose setae.

First pereopod (Figs. 7A and 7B): coxa naked, basis with 3–5 plumose setae; endopod sub-chelate, 5-segmented, ischium, merus and carpus with 8–9, 4–5 and 2–4 simple and plumose setae distributed as illustrated, propodus produced beyond half the length of the dactylus bearing 5–7 + 4–6 simple setae distributed as illustrated and 2 spines distally, dactylus with 2–4 simple setae and 2 spines distally; exopod unsegmented, bearing 2 subterminal and 4 terminal plumose setae; pleurobranch developed.

Second pereopod (Figs. 7C and 7D): coxa naked, basis with 2–3 plumose setae; endopod sub-chelate, 5-segmented, ischium, merus and carpus with 4–6, 4–5 and 3–4 simple and plumose setae distributed as illustrated, propodus produced half the length of the dactylus bearing 4–5 + 3–2 simple setae distributed as illustrated and 2 spines distally; dactylus with 3–5 simple setae and 2 spines distally; exopod unsegmented, bearing 2 subterminal and 4 terminal plumose setae; pleurobranch developed.

Third pereopod (Figs. 7E and 7F): coxa naked; basis with 1–2 plumose setae; ischium, merus, carpus and propodus with 1–3, 3–4, 2–3 and 4–5 simple and plumose setae arranged as figured, luminous organ present on carpus; dactylus with 3–5 simple setae and 2 spines distally; exopod unsegmented, bearing 2 subterminal and 4 terminal plumose setae; pleurobranch developed.

Fourth pereopod (Fig. 7G): coxa naked; basis with 2 plumose setae; ischium, merus, carpus and propodus with 2–3, 1–2, 2–3 and 5–6 simple and plumose setae arranged as figured, luminous organ present on carpus; dactylus with 3–4 simple setae and 1 spine distally; exopod unsegmented, bearing 2 subterminal and 4 terminal plumose setae; pleurobranch developed.

Fifth pereopod (Figs. 7H and 7I): coxa naked; basis with 1–2 plumose setae; ischium, merus and carpus with 1–2, 1–2 and 4–5 simple and plumose setae arranged as figured; propodus with 1 stout plumose seta at two-thirds of the length of the segment and 5–6 plumose and simple setae distributed, luminous organ present on carpus; dactylus with 1–2 simple setae subterminally and 4 stout simple setae distally; exopod short, measuring less than one-third of the length of the endopod, bearing 4 terminal plumose setae; pleurobranch developed.

Pleon (Figs. 1, 6A and 6D–6G): third to fifth pleomeres with a posteromedial spine, being the one on somite 3 strong, the one on somite 4 slightly upturned, and the one on somite 5 which also bears a lateral spine, acute; sixth pleomere with a pair of lateral spines.

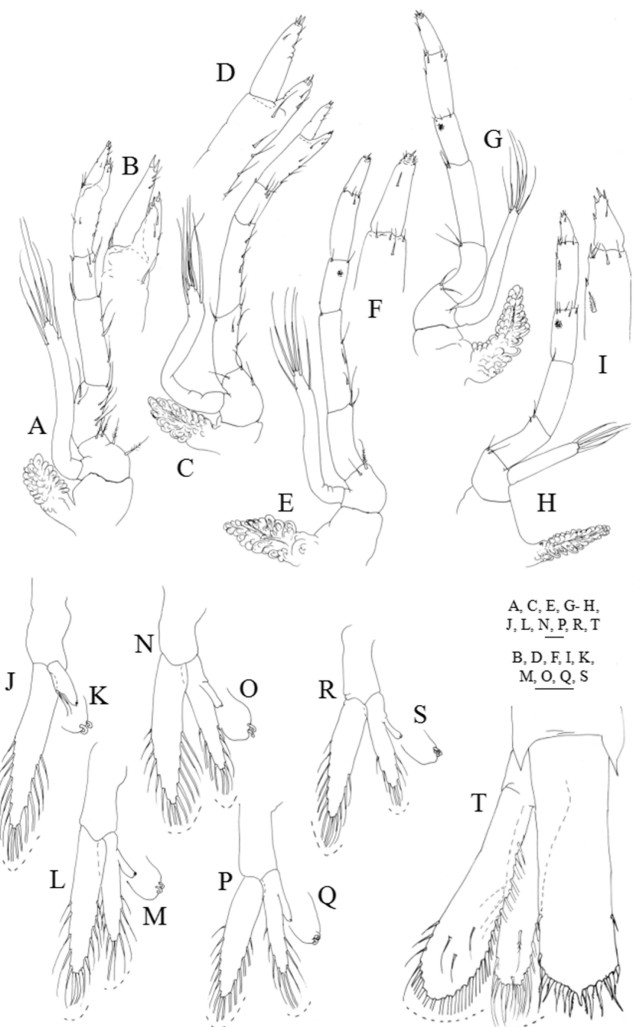

**Figure 7** *Systellaspis debilis,* **decapodid.** (A) First pereopod, (B) detail of propodus and dactylus of first pereopod, (C) second pereopod, (D) detail of propodus and dactylus of second pereopod, (E) third pereopod, (F) detail of propodus and dactylus of third pereopod, (G) fourth pereopod, (H) fifth pereopod, (I) detail of propodus and dactylus of fifth pereopod, (J) first pleopod, (K) first pleopod *appendix interna* posterior margin, detail of cincinulli, (L) second pleopod, (M) second pleopod *appendix interna* posterior margin, detail of cincinulli, (N) third pleopod, (O) third pleopod *appendix interna* posterior margin, detail of cincinulli, (P) fourth pleopod, (Q) fourth pleopod *appendix interna* posterior margin, detail of cincinulli, (R) fifth pleopod, (S) fifth pleopod *appendix interna* posterior margin, detail of cincinulli, (T) telson and uropods. Scale bars: 0.05 mm (K, M, O, Q, S); 0.1 mm (A–J, L, N, P, R, T).

Photophores present on the base of each pair of pleopods, close to their insertion in the pleon.

First pleopod (Figs. 6A, 7J and 7K): basipodite naked; endopod with 2 subterminal plumose setae and 2 cincinulli, exopod with 15–16 plumose setae.

Second pleopod (Figs. 6A, 7L and 7M): basipodite smooth; endopod with 10–11 plumose setae and the small *appendix interna* which presents 2 cincinulli, exopod with 16–17 plumose setae.

Third pleopod (Figs. 6A, 7N and 7O): basipodite smooth; endopod with 10 plumose setae and the small *appendix interna* which presents 2 cincinulli; exopod with 14–15 plumose setae.

Fourth pleopod (Figs. 6A, 7P and 7Q): basipodite smooth; endopod with 9–10 plumose setae and the small *appendix interna* which presents 2 cincinulli; exopod with 14–15 plumose setae.

Fifth pleopod (Figs. 6A, 7R and 7S): basipodite smooth; endopod with 8–9 and the small *appendix interna* which presents 2 cincinulli; exopod with 13–14 plumose setae.

Uropods (Fig. 7T): protopod unarmed; endopod with 19–20 plumose setae along inner and distal margins, 4–5 sparsely plumose setae along outer margin and 2–5 sparsely plumose setae distributed on ventral margin; exopod with 24–25 plumose setae along inner and distal margins, 2–3 sparsely plumose setae along outer margin and 2–4 sparsely plumose setae distributed on ventral margin.

Telson (Fig. 7T): parallel-sided, presents a slightly convex distal margin with a short median spine, two pairs of small lateral spines and six pairs of processes distally (the first pair is the shortest, the second pair the longest, and the second to the sixth are plumose); small anal spine now present.

## DISCUSSION

### Morphology of the larval cycle of *Systellaspis debilis* and larval characters for the Oplophoridae

The complete larval development of *Systellapis debilis* described in present study has four zoeae and one decapodid, adding a zoea to the until now known larval sequence for this species. The *Gurney & Lebour (1941)* description, based on a sequence of larval stages from plankton of Bermuda waters, presents a larval cycle of three zoea and a post-larval stage (=decapodid). However, *Gurney & Lebour (1941)* indicate that the second zoeal stage has a telson with 8 + 8 posterior processes and free uropods, whereas the results from this study shows that the uropods are visible through the telson integument but not free. In fact, *Gurney & Lebour (1941)* had concluded that the second zoea corresponded to a normal third stage. Nonetheless, after a detailed morphological comparison between present work and *Gurney & Lebour (1941)* *Systellaspis debilis* larval descriptions, we found only minor differences, which concerns the setation of some appendages and the dimensions of the larvae. When comparing the first zoea, the maxilla previously described lacks detail in the coxal and basial lobes with 2 + 0 and 1 + 1 small setae, which are ornamented with 2 + 2–3 and 2 + 3 setae in the coxa and basis, respectively. The telson is described as having the uropods under its integument, which in the present work are visible only in the second zoeal stage.

The second zoea described in present study was not previously observed by *Gurney & Lebour (1941)*. For the third zoeal stage, the only difference between the two descriptions is in the number of setae of the scaphognathite of the maxilla that presents 15–17 marginal plumose setae and 1 long distal stout setose process, instead of 13 marginal plumose processes as in *Gurney & Lebour (1941)* description. The fourth zoeal stage now presented

slightly differs from the *Gurney & Lebour (1941)* third stage, by the presence of two very small ventral teeth in the rostrum, and the more setae found in the maxillule and in the maxilla. *Gurney & Lebour (1941)* also compare the third larval stage with a specimen taken at Bermuda, considering that probably it is a fourth stage. Comparing this specimen with the fourth zoeal stage obtained in present study, we conclude that it differs in the flagellum of the antenna, which is longer than the exopod, and in the telson that is slightly wider at the posterior end, with a straight posterior margin (*Gurney & Lebour, 1941*).

We agree with *Gurney & Lebour (1941)* probable explanation for the observations made that there is some variation in the number of stages after the second zoea, also because the characters of the telson agree well with those described in present study for the fourth zoeal stage. Besides this, *Gurney & Lebour (1941)* describe another probable larval stage four, collected in the Discovery station 281 (see Figs. 6A–6C as in *Gurney & Lebour, 1941*), different from the one described in the present study for the decapodid stage. *Gurney & Lebour (1941)* stage four presents six ventral teeth in the rostrum, the carapace has an anterior ventral margin slightly serrated, the maxillule, maxilla and first maxilliped are figured, and the telson is described as more slender, presenting a long median spine. The authors then compare this larval form with the one described in *Kemp (1907)*, hypothesizing that both correspond to the same stage, possibly of a different species or of an intermediate stage between the last zoeal stage and the first decapodid. When compared with the fourth zoea described by us, they differ in the number of ventral teeth of the rostrum and in the form of the telson, which in the present study is rectangular shaped with a median spine with the same size of the other telson posterior processes. Regarding the endopod of the maxillule of the specimen collected in the Discovery station 281 (*Gurney & Lebour, 1941*), it seems unsegmented, which is segmented in the fourth zoea described here; also, the endopod of the maxilla possess only 3 setae which in the present description has 2 + 1 + 1 + 3 plumose setae. The larvae described by *Gurney & Lebour (1941)* presents a first maxilliped similar to the one observed for a decapodid stage, with a 3-segmented endopod, the exopod with plumose setae along the outer margin, and the epipod bilobed, whereas the fourth zoea described here presents a first maxilliped typical of a zoeal stage, with a 4-segmented endopod, the exopod with plumose setae terminally and a small epipod. Therefore, since the Discovery station 281 larvae described by *Gurney & Lebour (1941)* has a telson and the first maxilliped similar to those of the decapodid stage, and the endopods of both, the maxillule and maxilla, are different in shape and number of setae, we consider that, probably, it belongs to a different Oplophorid species.

It is recognized that the number of stages can be a consequence of the genetic differences between regionally separated populations (e.g., *Anger, 2001*), of the energy content of larvae at hatching as a proxy for the per offspring investment (e.g., *Oliphant & Thatje, 2013*), or of the environmental conditions such as the temperature (e.g., *Oliphant, Hauton & Thatje, 2013*). The plasticity in the number of larval stages is common for shrimps, and results published to date show that temperature, salinity and food affect the moult cycle, and can produce intermediate forms (e.g., different morphs of *Nauticaris magellanica* from the third zoeal stage on, as described in *Wehrtmann & Albornoz (2003)*). Likewise, besides the morphological plasticity, different populations of the same species

can have larval cycles with a different number of stages (e.g., populations of *N. magellanica* from the Argentine waters- Atlantic ocean- and the Chilean waters- Pacific ocean- differed in the number of stages, size, setation of thoracopods and development of pereopods, as described in *Thatje & Bacardit (2000)*). Back to 1941, *Gurney & Lebour (1941)* were already aware of this, indicating that the differences observed between the Bermuda and the Discovery specimens might be due to the individual variation in the degree of development between the two populations. This plasticity is more common for long larval series than for short ones (e.g., *Anger, 2001*), where the critical periods related with quantity and quality of energy affect the moult cycle (e.g., *Wehrtmann & Albornoz, 2003*). The larval cycle of *Systellaspis debilis* has been shown here to be fully lecitorophic, and possibly less variation is expected for this group of species, since the changes in the energy requirements throughout development are already provided in the strongly enhanced yolk stores.

Gurney's decapodid, described as "post-larval 1" (*Gurney & Lebour, 1941*) is smaller in size when compared with those obtained in present study, but shows a longer rostrum with thirteen dorsal teeth and eight small ventral teeth and, the telson illustrated seems to be more developed. On the other hand, *Coutiére (1906)* presented a larval form that can be considered as a decapodid stage, having a TL of 11 mm (see Fig. 2A, in *Coutiére (1906)*), the rostrum shorter than the scaphocerite, with the same number of dorsal teeth and a smaller number of ventral teeth (13 and 3, respectively). The distribution of the luminous organs and of the photophores is in agreement in all three descriptions. Both authors (*Coutiére, 1906*; *Gurney & Lebour, 1941*) mention that the pereon is full of yolk, as observed in the present study.

*Gurney & Lebour (1941)* stated that the morphological larval characters for *Systellaspis* and *Oplophorus* are very similar, reflecting a closer relation between both genera. Besides the presence of the luminous organs, the two larval series differ on the antennal scaphocerite which in *O. spinosus* has a stout apical spine from first stage on, a small exopod on pereopod 5, and a reduced ocular pappila. It is also stated, that probably the mouthparts remain not functional due to the mass of yolk observed in the body (*Gurney & Lebour, 1941*), an assumption in agreement with our observations. Considering present knowledge, we conclude that species of the family Oplophoridae will, most probably, have a larval cycle of four to five zoeae. The first zoea will have the exopod of the antenna unsegmented, will present buds for all pereopods and pleopods, which will be biramous in all pereopods, the pereon will be full of yolk through all larval development. The telson is broad presenting a small median spine on posterior margin, in the last stage of zoea.

## Lecithotrophy in *Systellaspis debilis* larvae and its implications on the ecology and evolution of oplophorids

We propose that the larval phase of *Systellaspis debilis* is full lecithotrophic, due to the findings within this study showing that the complete larval development, from the first zoea to decapodid, and first juvenile stage, occurred without feeding. Full lecithotrophy has been described as a probable evolutionary consequence of living in a habitat with ecological and physiological constraints (e.g., *Anger, 2001*), similar to the one where this

species lives. This is mostly known for decapods inhabiting extreme habitats such as land locked species (e.g., *Benzie, 1982* for *Caridina mccullochi*; *Couret & Wong, 1978* for *Halocaridina rubra*; and *Rodríguez & Cuesta, 2011* for *Dugastella valentina*, three atyid shrimps, respectively, with five, four and two non feeding zoeal stages; and *González-Gordillo, Anger & Schubart, 2010* for *Sesarma windsor* and *Metopaulias depressus*, two sesarmid crabs with two non feeding zoeal stages). For species living in extreme habitats the evolutionary selection pressures are related with scarcity of food, temperature variation and high salinity gradients. In some cases subsequent to the full lecithotrophy in one or more larval stages, facultative lecithotrophy is possible due to the quantity of yolk still stored in the pereon (e.g., *Anger, 2001*). A clear example of this pattern was described for a freshwater atyid with abbreviated development and parental care, *D. valentina* (Crustacea, Decapoda, Caridea), whose first, second and third juvenile stages are facultative lecithotrophs (*Rodríguez & Cuesta, 2011*). Observing the pereon of the juvenile stage of *Systellaspis debilis*, is possible to state that it is full of yolk, leading to the supposition that maybe, similarly to what was previously observed in the obligate lecithotroph *D. valentina*, also *Systellaspis debilis* juveniles could be facultative lecithotrophs (*Rodríguez & Cuesta, 2011*).

*Gurney & Lebour (1941)* separated the oplophorids in two distinct groups according to the size of the eggs: the large eggs genera *Ephyrina*, *Hymenodora*, *Oplophorus* and *Systellaspis*, vs the small eggs genera *Acanthephyra* and *Notostomus*. The authors related the size of the eggs with the number of zoeal stages in the larval cycle predicting that species with small eggs would have a long larval cycle, and the ones with large eggs will have a short larval cycle of no more than five zoeal stages (*Gurney & Lebour, 1941*). The females of *Systellaspis debilis* observed in the present study carried between 2 and 12 very large eggs (Table 1), supporting that prediction. Consequently, knowing that the species *Hymenodora glacialis* and *O. spinosus* besides the size of the eggs shared other characteristics such as the presence of pereopods and pleopods from the first zoeal stage (*Gurney & Lebour, 1941*), we hypothesize that most probably these species will have a fully lecithotrophic larval development as well.

*Gaten & Herring (1995)* hypothesized that the larvae of species of the genera *Oplophorus*, *Systellaspis*, *Ephyrina* and *Hymenodora*, with large eggs (*k*-strategists), showing an extended parental care or enhanced female energy investment in a reduced number of offspring, occur at depths of 200 m or more. Contrasting with these, the species of the genus *Acanthephyra*, produce large numbers of larvae (*r*-strategists, investing in a high number of eggs), are distributed at depths of 200 m or upper. In fact, the larvae of *Systellaspis debilis* are rare in the plankton of the euphotic zone (up to 200 m depth) as shown in e.g., *Dos Santos (1999)*, *Torres et al. (2014)* and *Pochelon et al. (2017)*. Therefore, they probably inhabit in deeper layers of the water column, restricting their dispersion, skipping the need and search for food in an oligotrophic environment such as the mesopelagic zone. This might be a strategy acquired to enhance the survival of the species, stabilizing the evolutionary trajectory imposed and in accordance with the recent evidence of cryptic speciation in *Systellaspis debilis* (Atlantic vs. Pacific and Indian Oceans, as the recent results by *Lunina, Kulagin & Vereshchaka (2018)*, seem to demonstrate).

Considering that the phylogenetic reconstructions are very effective for tracing character evolution and major life-history adaptations, a greatly reduced dependence of a vulnerable free living larva on planktonic food sources, is treated as a derived character, i.e., an advanced trait in the evolution (e.g., *Anger, 2001*), and an increase in maternal investment is considered a crucial step in the shift to non-feeding lecithotrophic development (e.g., *Strathmann, 1978*). Considering the recent phylogenetic reconstructions, *Bracken et al. (2009)* suggested the polyphyly of the Oplophoridae, and *Chan et al. (2010)* recommended to consider the two families, Oplophoridae and Acanthephyridae. Later, *Wong et al. (2015)* phylogenetic analyses to trace the evolution of bioluminescence within the oplophorids, resulted in a two major clades topology, strongly supporting the monophyly of the first clade, the family Oplophoridae (Bayesian analysis—posterior probabilities and maximum likelihood—bootstrap values displayed of 100/100). The authors considered that the emergence of photophores, also occurred later in the evolution, and was restricted to the first clade. The lecithotrophy observed for the oplophorids seems to be similar to what has been stated for the sesarmid crab (e.g., *González-Gordillo, Anger & Schubart, 2010*), where the planktotrophy–lecithotrophy dichotomy appears to show the evolutionary radiation from the marine to the terrestrial environment, with the percentage of increased yolk storage, allowing, respectively, planktotrophy, facultative lecithotrophy or full lecithotrophy. Therefore, the oplophorid shrimps seem to be good candidates to study the evolutionary correlation between the two developmental pattern extremes in the mesopelagic environment: the planktotrophy in the larvae of the Acanthphyridae vs the full lecithotrophy of the Oplophoridae.

Enhanced yolk reserves constitute a strategy to reduce the nutritional vulnerability of the newly hatched 1arvae (e.g., *Anger, 2001*). Accordingly, considering a probable evolutionary trend toward a shortening of the planktotrophic larval phase (e.g., *Strathmann, 1978*), it is accepted that the tendency toward lecithotrophy is associated with a decrease in the number of larval stages (4–5 zoeal stages; e.g., *Anger, 2001*). An abbreviated larval development similar to the one observed for *Systellaspis debilis* in the present study was already recorded for the larvae of *Pasiphaea japonica* (family Pasiphaeidae; *Nanjo & Konishi, 2009*), hence both species hatch with the pereopods and pleopods as buds, and complete their larval cycle with four full lecithotrophic zoeal stages. But other developmental patterns are observed in the nature, describing different evolutionary trajectories. Recently, *Hernández-Ávila, Cambon-Bonavita & Pradillon (2015)*, while studying the larvae of the alvinocaridids, concluded that the observed undeveloped mouthparts and the large amount of lipid reserves strongly support the occurrence of primary lecithotrophy. Though, the alvinocaridids larvae undeveloped mouthparts are combined with the lack of pereopods and pleopods, suggesting lecithotrophy only in the first zoea, a stage that can last 70 days. The authors suppose that the combination of undeveloped mouthparts with the absence of pereopods and pleopods in the newly hatched zoeae, suggests lecithotrophy, in an extended larval development (with more that 4–5 stages), defining a life history model consistent with a wide dispersal strategy in the oligotrophic environment where these species live (hydrothermal vents and/ or cold seeps).

## CONCLUSIONS

Present work paramount the need for more larval descriptions of representatives of the superfamily Oplophoroidea, but also of other mesopelagic species, in order to elucidate the general patterns of development in its two families, understanding the taxonomic significance of different life history strategies and, clarifying the phylogeny of this group. The complete larval development of *Systellapis debilis* is described from laboratory reared material. The four zoeal and one decapodid stages were found to be full lecithotrophic. Considering the previous descriptions for the Oplophoridae, which larval cycle consists of four to five zoeae, we propose as larval characters for this family: the exopod of the antenna unsegmented from the first zoea, all pereopods and pleopods present as buds from the first zoea, exopods in all pereopods, the telson broad with a small median spine on posterior margin in the last zoeal stage, and the pereon full of yolk through all the larval development, suggesting a full lecithotrophic development. The full lecithotrophy pattern is a probable evolutionary consequence of living in the mesopelagic zone, and seems to be an ecological adaptation of this species to this extreme environment. Knowing that the larvae of *Systellaspis debilis* are rare in the plankton of the euphotic zone, they inhabit the deeper layers of the water column. Due to the yolk-based development, the larval survival is enhanced in the mesopelagic oligotrophic zone, as there is no search for food. Considering the phylogenetic analyses published to date and the observations made in present study, we consider that similarly to what has been stated for some land locked crabs, also the oplophorid shrimps, can be a good model to study the lecithotrophy vs planktothrophy extremes in the mesopelagic environment.

## ACKNOWLEDGEMENTS

The specimens used in present study were collected and reared under appropriate permits and approved ethics guidelines.

### Funding

Cátia Bartilotti was supported by Fundação para a Ciência e a Tecnologia (FCT) through the postdoctoral fellowship SFRH/BPD/63888/2009, as well as by postdoctoral fellowships in the projects "BIOMETORE—Biodiversity in seamounts: the Madeira-Tore and Great Meteor" (http://www.biometore.pt/) funded by PT02 program (EEA Grants), and "PLANTROF- Dinâmica do plâncton e transferência trófica: Biodiversidade e ecologia do zooplâncton de Portugal" funded by Mar 2020—Programa Operacional Mar 2020 Portaria N.° 118/2016. The ovigerous females of *Systellaspis debilis* were collected in two different research surveys financed by the projects MedEx (MARIN-ERA/MAR/0002/2008; CTM2008-04036-E/MAR; EC FP6 ERA-NET Programme), and Nephrops Survey Offshore Portugal (PNAB-DCR-Data Collection Regulation). The study contributes to EMODNET EASME/EMFF/2016/1.3.1.2 – Lot 5/SI2.750022 Biology project. The funders

had no role in study design, data collection and analysis, decision to publish, or preparation of the manuscript.

## Grant Disclosures

The following grant information was disclosed by the authors:

Fundação para a Ciência e a Tecnologia (FCT): SFRH/BPD/63888/2009.

BIOMETORE—Biodiversity in seamounts: the Madeira-Tore and Great Meteor: PT02 program (EEA Grants).

PLANTROF- Dinâmica do plâncton e transferência trófica: Biodiversidade e ecologia do zooplâncton de Portugal: Mar 2020—Programa Operacional Mar 2020 Portaria N.° 118/2016.

MedEx (MARIN-ERA/MAR/0002/2008; CTM2008-04036-E/MAR; EC FP6 ERA-NET Programme) and Nephrops Survey Offshore Portugal (PNAB-DCR-Data Collection Regulation).

EMODNET EASME/EMFF/2016/1.3.1.2 – Lot 5/SI2.750022 Biology.

## Competing Interests

Antonina Dos Santos is an Academic Editor for PeerJ.

## Author Contributions

- Cátia Bartilotti conceived and designed the experiments, performed the experiments, analyzed the data, contributed reagents/materials/analysis tools, prepared figures and/or tables, authored or reviewed drafts of the paper, approved the final draft.
- Antonina Dos Santos conceived and designed the experiments, performed the experiments, analyzed the data, contributed reagents/materials/analysis tools, prepared figures and/or tables, authored or reviewed drafts of the paper, approved the final draft.

## Field Study Permissions

The following information was supplied relating to field study approvals (i.e., approving body and any reference numbers):

The field experiments were approved by IPMA, and the specimens used in present study were collected and reared under appropriate permits and approved ethics guidelines. The research surveys were carried out by Instituto Português do Mar e da Atmosfera, I.P. (IPMA, former IPIMAR), in August 2010 (Oceanographic Survey MedEx- 02060810, coordinated by Antonina dos Santos, project MedEx funded by MARIN-ERA/MAR/0002/2008; CTM2008-04036-E/MAR; EC FP6 ERA-NET Programme), and in June 2011 (Nephrops Survey Offshore Portugal- 02030611, coordinated by Cristina Silva, project Nephrops Survey Offshore Portugal funded by PNAB-DCR- Data Collection Regulation).

## Data Availability

The sampled females are described in Table 1, and the code of the larval series is deposited in IPMA—Instituto Português do Mar e da Atmosfera, in Lisbon, Portugal, is IPIMAR/O/Sd/01/2011 (Materials and Methods). Measurements are in the Results section.

All the raw data is included in the article:

- information about the sampled females used in present study in Table 1, in the Results section

- number of larvae analyzed per stage in lines 146 (first zoea), 183 (second zoea), line 228 (third zoea), line 280 (fourth zoea) and line 356 (decapodid), in the Results section

- measurements taken (total length (TL), carapace length (CL) and rostrum length (RL)) in lines 146 (first zoea), 183 (second zoea), line 228 (third zoea), line 280 (fourth zoea) and line 356 (decapodid), in the Results section

- Figure 1 showing the yolk stored in the pereon of *Systellaspis debilis*, decapodid stage, in the Results section

- Figures 2–7 presenting the complete larval description of *Systellaspis debilis* (first to the fourth zoeal stage, and decapodid), in the Results section.

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
