# Peer review of "The secret life of deep-sea shrimps: ecological and evolutionary clues from the larval description of Systellaspis debilis (Caridea: Oplophoridae)"

_PeerJ, doi:10.7717/peerj.7334_

## Round 0.1 · original submission · Minor Revisions

The reviewers have provided some really helpful and detailed comments that should improve the quality of your interesting paper - I suggest that you adopt most of them. I was Ted Gaten's PhD student (you cite him in relation to eye development). We did some other work that may add more weight to your interesting suggestion that S. debilis larvae live in deeper layers of the water column. We used dorsal tapetal distribution (eyeshine) as a proxy for the preferred depths of mesopelagic species and when we compared S debilis, S cristata, A pupurea and O spinosus we found that only in S debilis did the tapetal distribution decrease dorsally with maturity. This means that their dorsal eyeshine was less and suggests that they had shallower distributions as adults than as larvae.

Johnson ML, Shelton PMJ, Gaten E, Herring P (2000) Relationship of dorsoventral eyeshine distributions to habitat depth and animal size in mesopelagic decapods. Biol Bull 199:6–13

Reviewer 1 ·

Basic reporting

The language should be improved throughout to ensure that the reader can clearly understand. The authors need to be more specific in details and avoid using vague or descriptive words.

Some words are missing from the sentences such as “to” and “the”.

The flow of the introduction and discussion seems disjointed and hard to follow.
The author needs to clearly indicate what the papers main aims are? Introductions have the main aim of showing the reader the problem and the proposed solution.

I would start with the problem: no or limited verified published information on larval development for Oplophorid shrimps other than plankton samples.

Solution: Describe the development from reared larvae. Why is this important: this can give information about the species ecology and evolutionary development. Add some background information surrounding larval development strategies.

The authors should give the reader the information of why was the investigation was carried out and why this species was specifically analysed?

I feel there is lost opportunity within the article to give valuable information on the specifics of the larviculture process. The egg dimensions and number hatching is reported in the methods rather than the results. No survival information is provided. The growth / size data could be provided in a table.

The article structure seems to be good with standard sections.

The figures are clear and presented well. However, the scale bars could be made clearer.

No raw data was attached. The table presented provides data on the number of broodstock, number of eggs and number of larvae. No data on the survival or growth for the larvae is provided.

Some citations do not follow PeerJ guidelines. Some references are not cited within the manuscript.

Specific Details:
Line 25: I recommend replacing “outburst” with “attention”.
Line 33-34: Seems like a stand-alone statement. Could you explain or go onto explain why this is important?
Line 35: The use of the Gurney as a reference throughout the manuscript doesn’t seem correct as not all the authors are being cited. See appropriate peerJ referencing styles. i.e. “Gurney & Lebour (1941) described the…”.
Line 42: Please re-phase. You could use “an” instead.
Line 44 – 46: This sentence is hard to follow. Please re-phase and be clear on your main point that the quantity of egg yolk is in some regards correlated to the larval feeding strategy, i.e. either lecithotrophic or planktotrophic. A simple explanation of what theses two strategies are should be provided and how this relates to the grouping of the species / split of the genera.
Line 47: A leading sentence is needed here to flow from larval feeding strategies to larval morphology. The main point is also missed: that within the supergroup there is a wide variety of zoeal development stages before metamorphosis.
Do you mean “Only a small quantity of the species, within the Oplophoroidea superfamily have the larval developmental stages described from samples which were collected from plankton tows. These were …”. This paragraph needs clarity in how it leads to the next paragraph. Can you explain why Systellaspis debilis was focused upon?
Line 60-62: Please re-phrase. I find that sentence hard to understand, between which two stages? The authors should clearly state that there is confusion as to whether there is another stage before metamorphosis.
Line 67- 69: The finding that larvae are full lecithotrophic should be moved to within the results section and not shown within the introduction unless there is background knowledge corresponding to this?
Line 74: Please give the full name before using acronyms. i.e. Instituto Português do Mar e da Atmosfera (IPMA).
Line 80: Was there any cleaning of the eggs prior to transfer? Can you explain in the methods how you measured the eggs, for example longest width x length.
Line 80: The table should be presented within the results.
Line 82-83: “hatching that in some cases occurred almost one and a half month after collection” Should be in the results section. Could say instead along the lines of ”for a maximum of .. days?”.
Line 84: please be more specific as to what the larvae were transferred to. A 30 ml glass beaker? Information on the larval conditions should be indicated. For example, Was the conditions similar to that of the eggs - kept at a salinity of 35± 1, a temperature of 18+ 1 ºC, and a photoperiod of 12 h light: 12 h dark? Please clarify. Was the water replaced with autoclaved or filtered SW. If it was filtered than more information should be given.
Line 86: Can you give information on the Artemia sp. and how it was prepared, for e.g. similar to the methods used by ….
Line 88-90: Could be more succinct. “When the larvae reached the decapodid stage they were fixed in 4% borax buffered formaldehyde along with the exuviae for later morphological analysis.”
Line 98: “is referred to”
Line 115: An average should be accompanied with SD or SE.
Line 139 and throughout: I would avoid the word “very” or “little” as it seems too descriptive or ambiguous.
Line 166 and throughout: Fig 3A & 3A1
Line 433: I’m not sure what you mean by sequence? DNA sequence / larvae stages / development? Please clarify.
Line 433-434: Please re-phase. . I would try to avoid the use of his, ours and we. Rather specifically refer to each. For example “However, Gurney & Lebour, (1941) indicates that the second zoea stage as … whereas the results from this study indicate that ….
Line 433-437: Please make this sentence easier to understand.
Line 442 & throughout: I would avoid using “are in fact”. Please re-phase.
Line 448: I would replace “now presented” with “in this study”.
Line 454: “when he says” – please clarify.
Line 454-456: This needs to be in a separate paragraph and expanded upon with the paragraph below.
Line 456 & 458 & throughout: “by the present”
Line 465: give example of some of the decapods that show developmental stage plasticity.
Line 465-466: please re-phase. For e.g. Gurney … indicated that the differences observed
Line 467: Please clarify what you mean by “individual variation in the degree of development”
Line 468: Please re-phase. Be more specific, for e.g. “The plasticity shown in the number of larval stages is more common…”
Line 469-470: Can you explain why the proposal of the species being full lecithotrophic would increase the possibility of plasticity in the number of larval stages?
Line 474 - 477: “a larve récemment éclose” I do not understand this. Is this comparative forth stage or decapodid? Please re-phase and indicate more focus to the reader what it is you trying to show.
Line 480: Remove Fig 1.
Line 480-481: This should be in a new paragraph. There needs to be a clear flow through the discussion with a sentence introduction to each new paragraph building on the focus of the discussion. For example. “We propose that S. debilis is a full lecithotrophic species due to the findings within this study showing complete metamorphosis without the addition of food. Being lecithotrophic has been described as a probable evolutionary consequence of the habitat where this species inhabits (REF).“
Line 485: please re-phase sentence starting with “in that situations” as it is unclear.
Line 497: The sentence is a little confusing as you have said that S. debilis is full lecithotrophic. I think it would be beneficial to the reader to introduce the different energetic aspects of larval development within the introduction or the beginning of the discussion, highlighting the difference between planktotrophy, full, primary and secondary lecithotrophy. I would give further examples of similar species (if known) of each. In addition, do the authors assume that feeding will commence again at the onset of metamorphosis as direct observation of the development of the mouth parts?
Line 498: I am unsure of the use of the word "characters". Should this be characteristics? Or simply use "the larval morphology between … and .. are similar".
Lines: 498-499: Are you comparing all the species in two sets of genus? Should this not be Systllaspis spp. etc?
Line 500: Unsure of the word “series”. Should this be species?
Line 502: I think the readers may find it confusing when referring to “the author”. I think this should be changed to be specific. Which author are you referring to?
Line 512: general readers may not be familiar with the term “derived character”. This paragraph needs more of an introduction. It is unclear what the focus is for the reader and why the discussion has moved onto evolutionary development?
Line 514: the reference to the Fig 2, clade 1 is confusing. This should either be shown within the manuscript itself or the phylogenetic analyses clearly described to the reader.
Line 519: Please do not use the words actually and quite, and be specific. I am unsure what was observed? Please re-phrase this sentence.
Line 527-539: Similar to the earlier comment for line 497. The term “nutritional vulnerability” needs to be clearly explained and introduced earlier in the discussion. This may be a good paragraph to build upon to introduce the different energetic aspects of larval development.
Line 540-554: This paragraph could be made more succinct. It is unclear what you are trying to agree with or dispute? Please make this clearer and tie it in with the rest of the discussion.
Line 544: “more superficial depths” please be specific.
Line 546: there is a full stop between described and demonstrate. Please re-phrase the sentence.
Line 547: you refer to evidence but have not made clear or provided supporting results. It should be identified as speculation that the larvae reside within the mesopelagic zone and the argument given.
Line 551-554: Seems like stand-out sentences that need more explanation in relation to the studies aims and its implications, or it should be removed.
Line 571: I could not find a citation for this reference within the manuscript.
Line 614: I could not find a citation for this reference within the manuscript.
Figures: Really nice figures and clear drawings. Can the scale bars be relocated to avoid confusion and made clearer?
Within Figure 2 caption: C’ should be C1.
Within Figure 2 caption: Should A1 be included in the 0.05 mm scale? Please check all correspond to the correct scale.

Experimental design

The research question was not well stated nor was the potential implications of such research indicated and should be provided clearly and succinctly.
The morphological definitions and figures are of a high standard and are clear and precise.
More information is needed within the methods detailing the larval culture conditions to allow it to be reproducible by another investigator.
I think the manuscript would benefit with more information such as, egg development, hatch rate compared between years, larval survival data of each stage.
Yolk quantity or yolk nutritional quality using biochemical analysis would have been beneficial to the study.

I would have thought that one of the major questions of the research was the comparison between the two treatments /years, i.e. fed and un-fed? Factors such as survival / growth / microbiology (bacteria levels) to base the proposal of the larvae being lecithotrophic are not shown?
Was the feeding behaviour specifically observed for as mentioned within the text? If so this should be included in the methods. However, it would be hard to observe feeding behavior with live Artemia especially within the first developmental stages without the use of a microscope. What was the size / stage / nutrition of the Artemia as this can have an effect of the way larvae feed, at least for carnivorous larvae - for e.g. see Pochelon et al., 2009. Feeding Ability of early zoeal stages of the norway Lobster Nephrops norvegicus? Biol. Bull. 2016:335-343.
Was any quantification made on feeding rates i.e. the amount of feed provided compared to the amount of feed recovered to be certain that the larvae did not feed on the food provided?

Specific Details
Line 77: Is there any more information on the broodstock such as, what depth were they captured, what size, what moult period were they in?
Line 97: Similar to your earlier work and could be used as a reference: Bartilotti, Catia & Salabert, Joana & Dos Santos, Antonina. (2016). Complete larval development of Thor amboinensis (De Man, 1888) Decapoda: Thoridae) described from laboratory-reared material and identified by DNA barcoding. Zootaxa. 4066. 399. 10.11646/zootaxa.4066.4.3.
Line 110: Please clarify development until metamorphosis.

Validity of the findings

A fantastic achievement to culture species to metamorphosis for the first-time, identifying the species to be full lecithotrophic is a significant improvement to the knowledge gap for this species.

The discussion needs to have a clearer flow for the reader to follow and clearer focus on results related to the aims of the study.

The discussion seems disjointed with several topics discussed. The discussion starts by comparing the morphology in this species with that of other authors. The authors need to make the readers aware of the need to replicate of detailing the developmental stages of this species and the comparison with known descriptions from Gurney and Lebour (1941). The results indicate that are some differences and then indicate why this is important.

The issues of development stage plasticity needs to clearly augmented and the implications of this. More information indicating why this may occur should be included for example, different locations, depth, nutritional value within the yolk? The authors give no recommendation on further investigation, such as genetic analysis.

The proposal of the species being full lecithotrophic needs to fully proposed and augmented rather than stated as such. The proposal of the species being full lecithotrophic increasing the possibility of plasticity in the number of larval stages is not well explained or backed by any literature.

The discussion then jumps to the proposal that classifying larvae as lecithotrophic be based primarily on the size of the eggs is not well introduced to the reader and should be improved. Does this have any relation to the original aims of the paper?

The discussion then compares morphology of Systellaspis and Oplophorus, which I feel would be better suited to earlier in the discussion when comparing to Gurney and Lebour, (1941).

The proposal to study the evolutionary correlation between the two developmental pattern extremes in the mesopelagic environment between species is good but needs a clearer explanation and introduction.

The discussion then jumps back to the yolk reserves and the nutritional quality, although no specific details of nutrition were looked at within this study.

I feel from line 540 the discussion is not focused. The proposal that the species live in deeper layers of the water column needs a clearer explanation and linked with supporting results and/or literature.
The discussion needs to give the reader some implications of the findings and would benefit with a summary of conclusions in relation to the major findings of this paper.

Reviewer 2 ·

Basic reporting

I think this is a very nice piece of work. I have really enjoyed reading it.
This is a well-structured and well-written manuscript, authors did a great job. I think that is a detailed and complete study in a taxonomic group with few validated larval data, therefore this work provides relevant information.
I am not a native English speaker, therefore I have not tried to correct language, although I think it is correct.
I have only some minor comments, questions, and suggestions for helping to improve it.

Experimental design

Materials & Methods.
I would like to see some details about how eggs were conserved alive during cruises and transported to the laboratory. I mean, all larvae hatched at lab facilities? or in some cases some larvae were obtained onboard and later transfer to the laboratory on land? Also, I see that from females 1, 4, 8 and 9 you did not get any larvae, and in the case of female 10, just one from 11 eggs. They were not viable eggs? Or dead by….? I mean surely you get eggs in different degree of development, are there any relationship between eggs hatched and eggs not hatched with time of development? I see some eggs hatched after one month and a half of culture, so they were collected in early phase of development.

Respect to larvae hatched, I cannot see any comment about swimming behavior. These zoeae were active swimmers? In the figures of general view of zoeae I see short natatory setae, sound like not well swimmers.

In the description:
Line 96. Be more precise, “…long plumose setae on the exopods of maxillipeds and on…”, insert "of maxillipeds"
Line 110. Yolk was present in all stages, even decapodid, but what about first juvenile? According to amount of yolk in decapodid I guess that is still present in juvenile, and you obtained at least two juveniles (did you check this feature?) . In the case of Dugastella valentina, another caridean with abbreviated lecithotrophic development, yolk was observed until third juvenile stage (see Rodriguez & Cuesta, 2011).

Line 144. In the scaphognathite you describe 9-11 marginal plumose setae but in fig. 2F I see also a distal long stout process. I suggest to describe it.
Line 188. In the scaphognathite you describe 11-14 marginal plumose setae. But in fig. 3F I see only 10 marginal plumose setae and still a distal long stout setose process that you drawn perfectly differentiate of the rest of setae of this appendage. I suggest to describe it correctly.

In the setation of endopods of maxillipeds you describe all setae as plumose. I am not sure (based on your illustrations) that all of them are plumose, maybe some are simple setae? They look really small and without setules.

Lines 205-208. I agree that uropods are absent, but visible under integument in telson. Anyway, remember that uropods are appendages of sixth pleomere. I suggest to replace the sentence in telson to uropod description. As Uropods (Fig. 3K): absent, but already visible under telson integument.

Lines 232 and 287. The same that I comment above respect to distal long stout setose process still present in scaphognathite. It is just disappearing in the decapodid (fig. 6F).
Line 254.I cannot see in fig.4J that first pereiopod endopod is 5-segmented. I see lightly segmentation is marked, but not clear segments differentiated. And, of course this segmentation is more difficult to see in pereiopods 2-5.

Respect to zoea IV, you say in your description that pereiopods are all functional. What about maxillipeds are they functional as natatory appendages?? I mean, the pleopods are still not functional. How these larvae (zoea IV) move? “walking”? swimming?

Validity of the findings

I think that conclusions are based on data obtained and are well supported. I have only some comments for take into account in the Discussion section.

Line 443. Gurney describe the zoea I as having uropods “under skin”, you say that in the present work are not visible, but you could add “until zoea II”, where they are visible (fig. 3K).

Lines 483-487, and 519-523. I agree with citations here included, González-Gordillo et al. (2010) and Kattner et al. (2003), but they refer to brachyurans and lithodids, I suggest to check Rodriguez & Cuesta (2011), and references herein, where you can find examples of caridean (closer taxa) with direct, abbreviated and extended development (in this case evolutionary selection is not the same for all freshwater species). Also, you can find caridean larval descriptions (closer to Oplophoridae than brachyuran and lithodids) and see similar reduction of setation of mouthparts due to lecithotrophy adaptation. Maybe you can find more arguments to include in your discussion about larval lecithotrophy, mouthparts not functional, etc.

Rodríguez A & Cuesta JA (2011) Morphology of larval and first juvenile stages of the kangaroo shrimp Dugastella valentina (Crustacea, Decapoda, Caridea), a freshwater atyid with abbreviated development and parental care. Zootaxa, 2867: 43-58.

Additional comments

As mentioned above, I think this is a good work, therefore I have only comment some questions for helping you to improve it.

These are my last comments:
References
I check the manuscript and I cannot find the citation to dos Santos (1999) and Zariquiey-Alvarez (1968), I suggest to cite where appropriated or remove these references.

Figure captions
The figure captions are included twice, first from lines 617-66, and second from lines 668-717.
In figure 7, lines 662-664, the term “epipodite” is not appropriated used.

Table 1. I miss one date in this table, the size of the females. It would be nice to relate female size and number of eggs. If authors missed these data, maybe some comment in the Materials and Methods would be welcome. For example, the number of eggs was related with females´size or after fished the dead females could bear less eggs (loss during the process). I am curious about differences between females with just 2-3 eggs and other with 12 eggs.

·

Basic reporting

Overall: There are small grammatical and formatting errors throughout the manuscript. Please read carefully.

Many parts are well-written but there are areas of the manuscript where the English needs improvement. This is mostly the result of incorrect word choice that makes the manuscript hard to read in some areas. Please consult with a native English speaker to help with this part.

In some parts the text needs to be expanded.

The drawings are very nice!

Experimental design

There is some missing information on the experimental design. For example, I am confused by the addition of the 2010 study. What is the purpose of mentioning this study? Are results part of this manuscript?

Also, It is unclear “how many” individuals were examined for each stage. Please provide this information for each stage in the descriptions.

Other concerns are listed in the "general comments" below.

Validity of the findings

Overall fine, suggestions provided in general comments.

Additional comments

Overall: There are small grammatical and formatting errors throughout the manuscript. Please read carefully.

Many parts are well-written but there are areas of the manuscript where the English needs improvement. This is mostly the result of incorrect word choice that makes the manuscript hard to read in some areas. Please consult with a native English speaker to help with this part.

Title: I would consider shortening the title. It is a bit long winded for the description of the study.

Introduction:

Line 25: “recent outburst” is not the right use of the term here. Consider recent “attention”

The Wong et al. paper did not resurrect the family Acanthephyridae so please add the reference for this on line 33.

Overall the introduction is well written and important basic life history knowledge for mesopelagic shrimp. There are some additions I would like to see:

1. The introduction is very centered on oplophorid larvae, but what is missing is an overview on our state of knowledge for mesopelagic decapod shrimp as a whole. It would be great for the authors to “justify” why this work is important for deep sea biodiversity. This is not described in the paper much at all.
2. It was unclear how the present study differs from what is already known from Gurney and Lebour 1941. It needs to be emphasized how the present study enhances our knowledge on Systellaspis debilis and how it provides “better” or “more” knowledge then what is already in print. This is also not expanded well.

With these additions, the introduction will be a very nice contribution.

Methods:

The authors state in 2010 the shrimp did not eat Artemia. Is there are referenced study for this? If so, please add it.

I am curious about the lack of feeding. Is it possible that the larval development is “affected” by not eating? Has there been any studies that look at the effect of this in the literature? Some justification needs to be given as to how not eating should NOT affect the larval rearing process. One could assume as these shrimp are deep sea species, food would would be scarce anyways, but I would like to see some expansion here.

Results

I am very confused on why the authors are discussing the 2010 study. Is that part of this study? If so, this needs to be developed. It is unclear how the 2010 study fits into this study.

Also, the authors begin to make conclusion in the results section and this needs to be moving to the discussion. The results are to just report the results and not “discuss” the findings yet.

It is unclear “how many” individuals were examined for each stage. Please provide this information for each stage in the descriptions.

Discussion

I do not understand this sentence:
We agree with the author when he says that probably there is
455 some variation in the number of stages of the same species, since the characters presented for the
456 telson agree well with those described by present study for this larval stage.

Do the authors think there are possible more than 4 stages? Please revise to clarify.

The discussion on the two developmental patterns is very interesting and a valuable addition to the manuscript.

I would like to see the discussion dived into 2 sections: 1 that discusses the morphological characters and variability between studies and 1 that discusses lecithotrophy and the relation to evolutionary history.

The author do a very nice job comparing their findings with past literature on morphological differences between studies. It becomes a bit confusing what the final “conclusions” are concerning the variability between studies. At the end of section 1, I would like to see a conclusive paragraph that summarizes the major findings. For example, is it possible that the authors did not capture all the life stages? Or are they confident they described the full series and the other studies are incorrect? I was confused by all the different comparisons being made. The summary paragraph can summarize the findings of this study along with the others in a concise story.

Line 544: What are superficial depths? Acanthephyra spp. are hard to generalize as they display a WIDE distributional range with some species “stylorostratis” not migrating at all (Gulf of Mexico depth min of 750m) and others being shallow.

Line 544-547: the sentence is not complete. Please revise and describe the larval behavior.

I really appreciated the drawings in the manuscript. They look fantastic and I highly commend the authors for drawing the photophore pattern!!!!!

---

## Round 0.2 · Minor Revisions

Last time I looked at this paper I was under a lot of pressure from other things. I've had a bit more time this time and reviewing it again picked up on a few very minor grammatical errors and slightly confusing sentences. Also I have a suggestion for your table on egg characteristics. It will probably require less than an hour to make the changes suggested. See the attached pdf file. If you resubmit I will deal with the new version very quickly.

Reviewer 2 ·

Basic reporting

I have read carefully this revised version and compare it with the previous one. Also I have read the rebuttal letter by the authors. I think this is a clear improved version. I have no main corrections or comments.
Therefore, I think this manuscript version can be acepted in the current form for publishing.

Experimental design

It have been improved following some suggestions by referees.

Validity of the findings

The new discussion is improved with new data, comments and comparison with new citations. Findings as better supported now.

Additional comments

I think the revised version is clearly improved and deserve to be published.
I have found only a very minor thing to correct. In line 584 I think that references must be cited by year of publication, so "Dos Santos (1999), Torres et al. (2014) and Pochelon et al. (2017)", I do not understand the current order (based on...?).

---

## Round 0.3 · accepted · Accept

Thank you for your attention the the very minor amendments suggested in the last review. You might want to change the title to "The secret life of deep-sea shrimps: ecological and evolutionary clues from the larval description of Systellaspis debilis (Caridea: Oplophoridae)" and just take a look at the 1st sentence of the conclusion.